# Compounds Inhibiting Noppera-bo, a Glutathione *S*-transferase Involved in Insect Ecdysteroid Biosynthesis: Novel Insect Growth Regulators

**DOI:** 10.3390/biom13030461

**Published:** 2023-03-02

**Authors:** Kana Ebihara, Ryusuke Niwa

**Affiliations:** 1Degree Programs in Life and Earth Sciences, Graduate School of Science and Technology, University of Tsukuba, Tennodai 1-1-1, Tsukuba 305-8572, Ibaraki, Japan; 2Life Science Center for Survival Dynamics, Tsukuba Advanced Research Alliance (TARA), University of Tsukuba, Tennodai 1-1-1, Tsukuba 305-8577, Ibaraki, Japan

**Keywords:** glutathione *S*-transferases (GSTs), insecticide, insect growth regulator (IGR), mosquito, ecdysone, ecdysteroid, *Drosophila melanogaster*, *Aedes aegypti*, steroid, flavonoid

## Abstract

Glutathione *S*-transferases (GSTs) are conserved in a wide range of organisms, including insects. In 2014, an epsilon GST, known as Noppera-bo (Nobo), was shown to regulate the biosynthesis of ecdysteroid, the principal steroid hormone in insects. Studies on fruit flies, *Drosophila melanogaster*, and silkworms, *Bombyx mori*, demonstrated that loss-of-function mutants of *nobo* fail to synthesize ecdysteroid and die during development, consistent with the essential function of ecdysteroids in insect molting and metamorphosis. This genetic evidence suggests that chemical compounds that inhibit activity of Nobo could be insect growth regulators (IGRs) that kill insects by disrupting their molting and metamorphosis. In addition, because *nobo* is conserved only in Diptera and Lepidoptera, a Nobo inhibitor could be used to target IGRs in a narrow spectrum of insect taxa. Dipterans include mosquitoes, some of which are vectors of diseases such as malaria and dengue fever. Given that mosquito control is essential to reduce mosquito-borne diseases, new IGRs that specifically kill mosquito vectors are always in demand. We have addressed this issue by identifying and characterizing several chemical compounds that inhibit Nobo protein in both *D. melanogaster* and the yellow fever mosquito, *Aedes aegypti*. In this review, we summarize our findings from the search for Nobo inhibitors.

## 1. Introduction

Mosquitoes are responsible for the largest number of human deaths. They act as vectors for many pathogenic or parasitic infections such as malaria and dengue fever, which kill more than 700,000 people annually [1,2,3]. Therefore, it is essential to develop effective ways to control mosquito populations and prevent disease transmission. One method of mosquito control is to use insecticides that specifically act on cholinergic neurotransmission in insects and inhibit their motor functions. Organophosphates [4] and pyrethroids [5,6] are commonly used for mosquito control. However, some mosquitoes are developing resistance to them [7,8]. Therefore, instead of simply relying on a few insecticides, we must switch between multiple insecticides with different mechanisms of action and strategies to avoid the emergence of insecticide resistance. In this regard, the development of new insecticides with different chemical structures and mechanisms of action is required. Accordingly, the purpose of this review is to highlight an epsilon-class glutathione *S*-transferase (GST) in insects, the designated Noppera-bo (Nobo) [9,10,11], as a novel insecticide target.

## 2. Insect Growth Regulators (IGRs)

Insect growth regulators (IGRs) are drugs that exhibit high insecticidal or growth-inhibitory activity against insect pests, but have no effect on other organisms. IGRs cause death by disrupting insect molting and metamorphosis [12]. Some of these target insect-specific structural materials, such as chitins, which are essential for formation of the cuticular layer, and insect hormones, which are required for molting and metamorphosis [13,14].

The Insect Resistance Action Committee (IRAC) classifies 34 groups of insecticides based on mode-of-action, identifying the following groups as IGRs.

(1) ”Juvenile hormone mimics” (Group 7) regulate insect molting and metamorphosis by agonizing the action of juvenile hormones (JHs) and indispensable sesquiterpenoid hormones. Commercially used mimics include Methoprene [15,16,17,18], Fenoxycarb [19], and Pyriproxyfen [20,21,22]. These insecticides are effective against a variety of insect species, such as those belonging to the orders Diptera, Lepidoptera, Coleoptera, Orthoptera, and Hymenoptera [23].

(2) “Mite growth inhibitors affecting CHS1” (Group 10) [24], “Inhibitors of chitin biosynthesis affecting CHS1” (Group 15) [17], and “Inhibitors of chitin biosynthesis, type1” (Group 16) [25] inhibit chitin synthase 1 (CHS1) in the epidermis and prevent insects from molting, which leads to death. The CHS1 gene, which encodes enzymes that produce chitin in the cuticle, is found in the exoskeleton, and these insecticides target agricultural pests, such as mosquitoes, and mites.

(3) “Inhibitors of acetyl CoA carboxylase” (Group 23) disrupt the first reaction of lipid biosynthesis by inhibiting the enzyme, acetyl CoA carboxylase, resulting in death. The representative IGR, Spirodiclofen [26,27], developed as an acaricide, effectively controls a broad spectrum of sucking pests such as mosquitoes and mites.

(4) “Molting disruptor, Dipteran” (Group 17) causes irregular melanization and sclerotization of the cuticle, resulting in necrotic lesions, rupture of the insect body, and death in dipteran larvae. In this class, Cyromazine [28] is a widely used IGR, for which the target molecule is unknown.

(5) “Ecdysone receptor agonists” (Group 18) bind to the Ecdysone receptor, the nuclear receptor for active ecdysteroids that are indispensable in arthropods, including insects, but do not have significant physiological effects on other organisms [29]. Dibenzoylhydrazines are well-known commercially available ecdysone receptor agonists [14,30,31,32]. Dibenzoyl hydrazines include Tebufenozide, Methoxyfenozide, and Chromafenozide, which exhibit selective insecticidal activity against lepidopterans [32]. Although EcR is highly conserved in insects, the insecticidal activity of the Ecdysone receptor agonists is particularly high in lepidopteran insects, but weak in dipteran and coleopteran insects [33,34,35]. The emergence of resistance to these compounds has been reported [36].

The target molecule of IGRs is specific to insects, resulting in lower toxicity to non-target other animals, including humans, compared to other insecticides, thereby minimizing their impact on the environment [37]. IGRs classified in (2), (3), and (4) inhibit in vivo biosynthetic reactions through several mechanisms, while IGRs classified in (1) and (5) impact the binding of insect hormones, JHs and ecdysteroids by disrupting their receptors. However, no IGRs have yet been developed that target the biosynthesis of these hormones. Fortunately, in the past two decades, the biosynthetic pathways of JH and ecdysteroids and the enzymes involved have been identified and well characterized [38,39,40]. Therefore, IGR strategies that inhibit biosynthetic pathways rather than receptors are currently being considered [41]. In this review article, we focus on ecdysteroids, which are among the most crucial insect hormones.

## 3. Noppera-bo, the Ecdysteroidogenic GST

Ecdysteroids are arthropod steroid hormones, such as ecdysone and the more active 20-hydroxyecdysone, which regulate developmental and physiological processes in in-sects. Previous studies have shown that the amounts of active ecdysteroids, particularly 20E, in hemolymph fluctuate temporally during development [42]. Typically, surges of active ecdysteroid amounts are observed multiple times during development, which trigger temporal transitions in development, such as hatching, molting, metamorphosis, and eclosion.

During development, ecdysteroids are synthesized in the prothoracic gland (PG) from dietary cholesterol and plant sterol through multiple steps with various enzymes [39,43]. Ecdysteroid biosynthetic enzymes, including Nobo, collectively called Halloween enzymes [39], have been identified in *Drosophila melanogaster* and other insects over the past two decades [9,10,11]. Others include Neverland [44,45], non-molting glossy/Shroud [46], Spook/CYP307A1 [47,48], Spookier/CYP307A2 [48], Phantom/CYP306A1 [49,50], Disembodied/CYP302A1 [51], Shadow/CYP315A1 [51], and Shade/CYP314A1 [52]. *Spook*, *spookier*, *phantom*, *disembodied*, *shadow*, and *shade* encode cytochrome P450 monooxygenases, while others encode non-P450 type enzymes. All Halloween genes, except *shadow*, are specifically expressed in ecdysteroidgenic biosynthetic organs, including the PG and adult ovary, while *shade* is expressed in many peripheral tissues [39]. Genetic analyses of Halloween genes have been extensively conducted in *D. melanogaster*, and have revealed that the loss of Halloween genes causes embryonic or larval lethality due to defective ecdysteroid biosynthesis [53]. In the silkworm—*Bombyx mori*, a lepidopteran model insect—genetic mutants of *nobo* and *non-molting glossy*/*shroud* have been established. These also exhibit larval molting defects owing to the failure of ecdysteroid biosynthesis [11,46].

Among Halloween enzymes, Nobo is unique because it is the only ecdysteroidogenic enzyme belonging to the GST family, which are enzymes conserved in a wide range of organisms, from plants to animals [54,55,56]. GSTs generally catalyze conjugation of reduced glutathione (GSH) to substrates. Of the insect GSTs classified into six groups (delta, epsilon, sigma, theta, pi, and zeta) [57,58,59,60], Nobo belongs to the epsilon class. in vitro biochemical analysis has demonstrated that *D. melanogaster* Nobo (DmNobo; also known as GSTe14) exhibits steroid double-bond isomerase activity; however, corresponding steroid isomerization in vivo is unknown [61]. Although an endogenous substrate of Nobo has not yet been identified, genetic evidence strongly suggests that Nobo participates in cholesterol transport and/or metabolism [9,10,11]. Moreover, the requirement for GSH in ecdysteoid biosynthesis has also been confirmed; this is because a loss-of-function mutant of *glutamate-cysteine ligase catalytic subunit* (*gclc*), the critical enzyme for GSH biosynthesis, results in developmental lethality, partly due to a loss of ecdysteroid biosynthesis in *D. melanogaster* [62]. Thus, it is clear that GSH-dependent biochemical reactions mediated by Nobo are crucial for ecdysteroid biosynthesis in both dipteran and lepidopteran species. Of note, the developmental arrest phenotype observed in the loss-of-function mutant of *nobo* can be rescued by overexpression of *nobo* orthologs but not by the overexpression of other GST genes that do not belong to the Nobo family [10]. These observations suggest that, while most epsilon classes of GSTs have been reported to play a role in detoxification [63], Nobo is a GST specialized for ecdysteroid biosynthesis. Therefore, Nobo can be a target for IGRs that inhibit ecdysteroid biosynthesis (Figure 1).

GSTs are also used as tags to target proteins in genetic engineering, facilitating mass culture and purification of soluble recombinant proteins using an *Escherichia coli* expression system [64,65,66]. This is also the case for Nobo, as recombinant Nobo proteins can be easily produced by the *E. coli* expression system and purified using a commercially available conventional GSH column and gel-filtration chromatography [67,68,69]. Moreover, a high-throughput in vitro Nobo enzyme activity assay system has been established, which is optimal for screening inhibitors [67]. Taking advantage of these technical advances, Nobo is an attractive target for selective insecticides that kill only dipterans or lepidopterans, including mosquitoes, but does not kill beneficial insects, such as honeybees and ladybugs (Figure 1). In the remainder of this review, we focus on recent advances in the identification and characterization of compounds that inhibit enzymatic activity of DmNobo and yellow fever mosquito (*Aedes aegypti*) Nobo (AeNobo; also known as GSTe8), also known as *GSTe14* and *GSTe8*, respectively, and we discuss the search for vector mosquito-specific IGRs.

## 4. High-Throughput In Vitro Screening of Nobo Catalytic Activity

In general, several colorimetric and fluorometric methods have been used to detect GST activity in vitro. Conventional and typical substrates for colorimetric and fluorometric methods are 1-Chloro-2,4-dinitrobenzene (CDNB) [70,71] and monochlorobimane [72,73], respectively. However, both methods have low sensitivity and are unsuitable for high-throughput screening [41]. To overcome this low sensitivity, a novel fluorescent substrate, 3,4-DNADCF, was developed [67], which fluoresces approximately 54-fold upon GSH conjugation in the presence of GST. The enzymatic assay with 3,4-DNADCF was feasible even at a 1000-fold lower concentration than that of CDNB. Moreover, the high *k_cat_*/*K_M_* for 3,4-DNADCF is advantageous for the sensitive detection of GST enzymatic activity [67].

A high-throughput screening was conducted using 3,4-DNADCF with DmNobo recombinant proteins and a library of 9600 small-molecule compounds to identify inhibitors of DmNobo [69], and several inhibitors were identified [68,69,74]. Interestingly, these inhibitors included three steroid compounds, one of which was the mammalian steroid hormone 17 β-Estradiol (EST) [67]. EST is a strong inhibitor of DmNobo with 50% inhibitory concentration (IC_50_) of 1.2 ± 0.1 μM; however, it does not inhibit human GST P1-1 [67]. Nonetheless, EST is an endocrine-disrupting chemical, which renders it difficult to use as an insecticide [75]. Therefore, EST serves as a model compound to clarify the general mode of action of DmNobo inhibitors.

## 5. 17β-Estradiol Inhibits *Drosophila melanogaster* Nobo Enzymatic Activity

In 2020, the crystal structure of DmNobo was reported by Bengt Mannervik’s group and our group. The Mannervik group solved the co-crystal structure of DmNobo with GSH and 2-methyl-2,4-pentanediol, originating from the crystallization mother liquor (Protein Data Bank (PDB):6T2T) [61]. Meanwhile, our group solved crystal structures of DmNobo: the apo form of DmNobo (PDB:6KEL), co-crystals with GSH (PDB:6KEN), EST (DmNobo-EST; PDB:6KEO), and both GSH and EST (PDB:6KEP) [68]. For any of these structures, like other GSTs generally, DmNobo forms a homodimer, conserving the GSH-binding pocket (G-site) and hydrophobic substrate-binding pocket (H-site). As expected, GSH is intercalated into the G-site, and EST and 2-methyl-2,4-pentanediol were bound to the H-sites.

Several amino acids in DmNobo interact with EST and GSH, via hydrophobic interaction with Phe39. However, the most important of these is the aspartate residue located at position 113 (Asp113), which is situated in the innermost region of the H-site (Figure 2A and Table 1). Specifically, the Oδ atom of Asp113 forms a hydrogen bond with the O3 atom in the hydroxyl group of EST [68]. The importance of the hydrogen bond between Asp113 and EST is also indicated by the fragment molecular orbital (FMO) method, which evaluates inter-fragment interaction energy [76,77,78]. Moreover, a point mutant DmNobo protein, in which Asp113 is replaced by alanine, has no loss of enzymatic activity even in the presence of 25 μM EST, providing bio-chemical evidence for the importance of hydrogen bonding between Asp113 and EST.

The essentiality of Asp113 during *D. melanogaster* development has also been investigated using molecular genetics. A *D. melanogaster nobo* mutant allele carrying a D113A point mutation (*nobo^Asp113Ala^*) was generated using CRISPR-Cas9-based knock-in. *Nobo^Asp113Ala^* mutant animals exhibited embryonic lethality, characterized by a naked cuticle structure and failure of head involution [68]. These phenotypes are very similar to features of Halloween mutants, which exhibit a complete loss of *nobo* function [9,10]. The level of mutated DmNobo protein from the *nobo^Asp113Ala^* allele was comparable to that of wild-type DmNobo protein in the prothoracic glands, indicating that the Asp113Ala mutation did not affect the stability of Nobo. Asp113 of Nobo sustains the intrinsic biochemical function of Nobo and regulates its normal development in *D. melanogaster*.

Notably, Asp113 is almost perfectly conserved among six dipteran and 13 lepidopteran species with Nobo sequences in public databases [68]. This observation is consistent with the functional importance of Asp113 in Nobo in vivo. The only exception among Nobo proteins is AeNobo, which has Glu113 instead of Asp113 in DmNobo. However, hydrogen bond interactions between Glu113 of AeNobo and some inhibitor compounds were observed.

## 6. Flavonoidal Compounds, Such as Desmethylglycitein, Inhibit *Aedes aegypti* Nobo Enzymatic Activity

After identifying EST as a DmNobo inhibitor, it was initially speculated that EST would also inhibit the enzymatic activity of AeNobo derived from the yellow fever mosquito, *A. aegypti*. However, this is not the case. The IC_50_ of EST against AeNobo is 21.3 μM, which is approximately 10-fold higher than against DmNobo [69]. Therefore, to identify inhibitors of AeNobo, a high-throughput screening against AeNobo recombinant protein was performed using a library of 9600 small molecule compounds [69]. The compound, 2′-hydroxyflavanone, was identified with an IC_50_ of 4.76 μM. The inhibitory activities of other subclasses of flavonoids, including flavanone, flavone, isoflavone, flavanol, isoflavan, and anthocyanidin, were also tested against AeNobo. More than half of the tested flavonoid compounds exhibited inhibitory activity against AeNobo with IC_50_s less than 10 μM.

Three complex structures with three flavonoid inhibitors, daidzein (PDB:7EBU), luteolin (PDB:7EBV), and desmethylglycitein (DMG; IUPAC name:4′,6,7-trihy-droxyisoflavone; PDB:7EBW), were determined by X-ray crystallography. All three compounds formed hydrogen bonds with Glu113 in AeNobo (Figure 2B and Table 1 for DMG). Moreover, as in the case of DmNobo and EST, the enzymatic activity of a point mutation of the AeNobo protein substituting Glu113 with alanine was not inhibited by any of these flavonoids, even at a concentration of 25 μM. These results suggest that AeNobo Glu113 is essential for the inhibitory activity of flavonoids, as with Asp113.

Among the three flavonoids, DMG exhibits the strongest inhibitory activity against AeNobo, as the IC_50_ values of daidzein, luteolin, and DMG against AeNobo are 3.87 μM, 3.99 μM, and 0.287 μM, respectively. Although all of them form hydrogen bonds with Glu113 of Aenobo, the hydrogen bond itself cannot account for the difference in inhibitory activity between DMG and the other two. Our analysis suggests that a hydrophobic interaction between DMG and Phe39 of AeNobo is crucial for the higher inhibitory activity of DMG (Figure 2B and Table 1). Notably, the FMO calculation indicated a hydrophobic interaction between Phe39 of DmNobo and EST [61] (Figure 2A and Table 1).

A previous study reported that daidzein exhibited larvicidal activity against *A. aegypti* [79]. As DMG has stronger inhibitory activity against AeNobo than daidzein, it is expected to be a more effective larvicidal reagent. Indeed, this is the case, as the 50% leathal dose (LD_50_) of DMG was approximately 9.39 ppm, whereas daidzein exhibited an LD_50_ of 85.8 ppm. These results suggest that inhibitory activities of flavonoids are correlated with their larvicidal activity. In addition, DMG-treated animals (2.5 ppm) exhibited growth delays and suppressed expression of the ecdysteroid-inducible gene, *E74B*, consistent with our expectation that DMG inhibits the ecdysteoidogenic GST Nobo. This is the first report of a flavonoid compound potentially inhibiting the ecdysteroid biosynthesis pathway.

Many flavonoid compounds, including daidzein, have estrogenic activity and activate estrogen receptors [80,81]. Estrogenic compounds, including daidzein, have been recognized as dangerous endocrine-disrupting chemicals that can harm other animals [75]; therefore, they cannot be used as insecticides. Interestingly, a previous study showed that DMG lacks estrogenic activity [81], suggesting that it may not be an endocrine disruptor like EST or daidzein. However, DMG is less active than commercially available insecticides that effectively control mosquitoes. Nevertheless, we expect that DMG may serve as a good starting point for the development of highly active and environmentally friendly IGRs.

## 7. Glutathionylation of DmNobo Inhibitors

From our studies on DmNobo-EST and AeNobo-flavonoid interactions, hydrogen bonds formed with Asp/Glu113 and a hydrophobic interaction with Phe39 are the important features to achieve high inhibitory activity against Nobo. However, our investigation revealed that neither of these interactions are an absolute prerequisite for inhibitory activity.

Inhibitors of DmNobo include not only EST, but also five non-steroidal compounds [74]: TDP011 (IUPAC name:4-bromo-2-[4-(3-methoxyphenyl)-2,2-dimethyl-5,6-dihydro-1H-pyrimidin-6-yl]phenol), TDP012 (IUPAC name:1-(4-fluorobenzyl)-1H-thieno [3,2-c][1,2]thiazin-4(3H)-one 2,2-dioxide), TDP013 (IUPAC name:2-(5-tert-butyl-2-methyl-benzenesulfonylamino)-benzoic acid), TDP015 (IUPAC name:6-(6,7-dihydrothieno [3,2-c]pyridin-5(4H)-yl)-3-pyridinamine), and TDP044 (IUPAC name:2,2′-(1,1-ethanediyl)bis(3-hydroxy-5,5-dimethyl-2-cyclohexen-1-one). Interactions between DmNobo and these five inhibitory compounds were investigated by X-ray crystallography, and three-dimensional structures of co-crystals of DmNobo with GSH and TDP011 (PDB:7BD3), TDP012 (PDB:7DB4), TDP013 (PDB:7DAY), and TDP015 (PDB:7DAZ) have been determined.

All four chemicals insert into the H-sites of DmNobo. Surprisingly, unlike EST, none of the four compounds form a hydrogen bond with Asp113 (Figure 2C and Table 1 for TDP011). Moreover, none of the four compounds interacted with Phe39. Instead, the position of Phe39 changed between the apo-form and the complex forms with these non-steroidal compounds. When they are present in the H-sites, the aromatic ring of Phe39 moves away from the compounds, preventing their interaction with Phe39 (Figure 2C and Table 1 for TDP011). These data imply that the inhibitory activity of these non-steroidal compounds is achieved by another mechanism.

Interestingly, TDP011, TDP012, and TDP015 (but not TDP013) intercalate into the H-site only when GSH is present at the G-site. Moreover, these three compounds bind covalently to GSH in DmNobo co-crystals, indicating that TDP011, TDP12, and TDP015 are glutathionylated in DmNobo proteins. Glutathionylation is a chemical reaction in which a covalent bond is formed between the thiol group of the cysteine of GSH and the substrate. Notably, covalent bonds are stronger than the chemical bonds often used in human drug development. These drugs, called covalent drugs, bind irreversibly to their target proteins and exhibit potent, sustained effects [82]. Accordingly, we now speculate that such glutathionylation may be indispensable for some compounds to exhibit inhibitory activity against Nobo, but further studies are needed to clarify this point. If this is true, the identification and characterization of such covalent “IGRs” against Nobo may be a new direction to develop more potent Nobo inhibitors. However, covalent drugs may react nonspecifically with off-target proteins or biological components, causing side effects. Therefore, it may be crucial to select chemical reactions that exhibit high selectivity for Nobo proteins in vivo.

## 8. Conclusions and Future Perspectives

We summarized ideas for a new IGR strategy against Nobo for insect GSTs. In our previous studies on the structure of Nobo inhibitors [68,69,74], three key features were identified: (i) hydrogen bonds formed with Asp/Glu113, (ii) a hydrophobic interaction with Phe39, and (iii) glutathionylation of compounds. Currently known Nobo inhibitors exhibit one or two of these features. To create a more potent Nobo inhibitor, we propose that a stronger Nobo inhibitor can be developed by satisfying all three of these criteria. Currently, we are working on the identification and characterization of such compounds. We believe that this approach has potential to yield effective Nobo inhibitors for use as IGRs.

A variety of compounds with different chemical structures have been identified as inhibitors of DmNobo and AeNobo. Although the overall protein structures of DmNobo and AeNobo are similar, several amino acids surrounding the H-site differ between the two. Consequently, some DmNobo inhibitors exhibit weak hydrophobic interactions with AeNobo, resulting in low inhibitory activity. However, the differences in inhibitory activity among insect species cannot be solely attributed to the amino acid compositions or the size of their H-sites. Thus, the cause of the variation in inhibitory activity remains unknown. According to IRAC, commercially available IGRs, such as JH mimics, inhibitors of chitin biosynthesis, inhibitors of acetyl CoA carboxylase, and ecdysone receptor agonists, face resistance from certain species [7,8,65]. Moreover, insects that have acquired stronger resistance to IGRs may emerge more frequently in the future. Therefore, it is desirable to continuously develop IGRs with various mechanisms of action and to rotate their use with current insecticides.

Molecularly targeted insecticides, such as Nobo inhibitors, represent a safer and eco-friendly alternative, given their well-understood mode of action. To apply Nobo inhibitors as insecticides, it is imperative to undertake toxicity evaluations on both target and non-target insects. Furthermore, at the molecular level, Nobo inhibitors demonstrate greater potency than conventional GST inhibitors [83]. However, unlike existing IGRs, currently identified Nobo inhibitors are still less toxic to mosquitoes. For example, the LD_50_ of DMG, which is currently the most potent Nobo inhibitor, is approximately 10 ppm against *A. aegypti* larvae [69]. In contrast, the LD_50_ values for organophosphates and pyrethroids against mosquitoes are typically less than 0.1 ppm [84,85]. Therefore, a further combined approach, incorporating both in vitro and in vivo structural analysis, is required for a Nobo inhibitor with higher IGR activity.

As an additional curiosity, it is reported that the expression of Halloween genes is suppressed when JH mimics are administered to ex vivo cultured PGs from *D. melanogaster* and *B. mori* [86]. JH hormone mimics have also been reported as inhibitors of mosquitoes [15,17,18,23]. It is unclear whether JH mimics inhibit the enzymatic activity of Nobo, but if they inhibit the expression of Halloween genes, it is likely that they also inhibit Nobo. Since JH mimics possess multiple methyl groups, they are expected to have more hydrophobic interactions when bound to Nobo. On the other hand, the binding mode of the JH mimics is unknown, since few structures of JH mimics have been reported.

Lastly, it is crucial to identify the endogenous substrate of Nobo to completely understand its in vivo mode of action. The ligand is suspected to be a steroid that interacts with Asp/Glu113 of Nobo through a hydrogen bond and may be displaced from the H-site by Nobo inhibitors. Further studies are necessary to understand the chemical properties of the ligand and the relationship between the ligand, inhibitors, and Nobo proteins. These studies are important for any future use of Nobo inhibitors as IGRs.

## Figures and Tables

**Figure 1 biomolecules-13-00461-f001:**
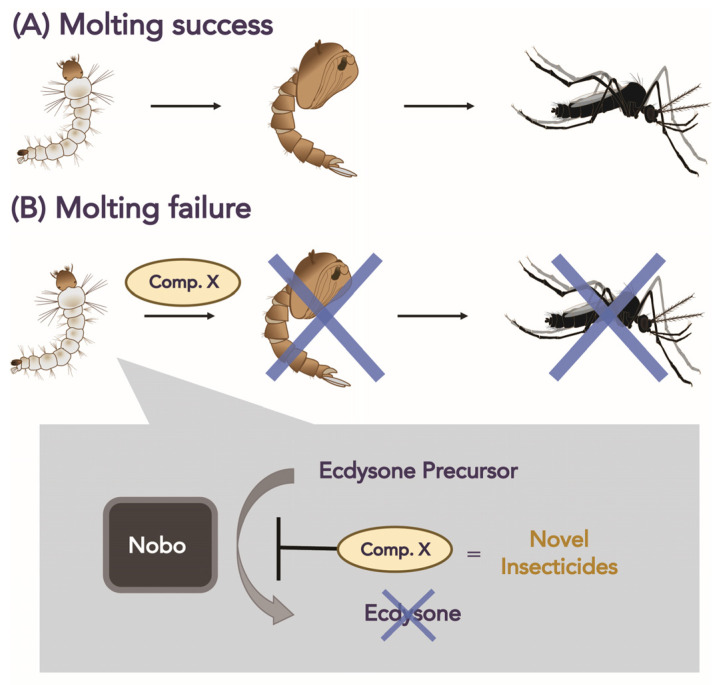
The action of a Nobo inhibitor affecting mosquito development. (**A**) Under normal conditions, mosquitoes are holometabolous insects, undergoing metamorphosis from larvae to pupae and finally to adulthood. Multiple molting stages occur during their development. (**B**) Exposure of early larvae to an inhibitor of Nobo (Comp. X) results in death prior to pupation. This phenomenon is attributed to the binding of Comp. X to Nobo, inhibiting ecdysteroid biosynthesis. Comp. X is anticipated to exhibit insecticidal efficacy as a novel IGR.

**Figure 2 biomolecules-13-00461-f002:**
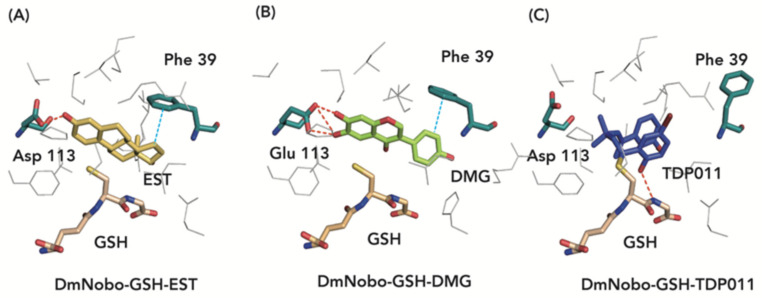
Characteristics of major Nobo inhibitors. Hydrogen bonds are represented by red lines, while hydrophobic interactions are represented by light blue lines. (**A**) EST (carbon atom depicted in yellow) interacts with DmNobo via Asp113 and forms a hydrogen bond. EST forms a hydrophobic interaction with Phe39. (**B**) DMG (carbon atom depicted in green) interacts with AeNobo via Glu113 and forms a hydrogen bond. DMG interacts hydrophobically with Phe39. (**C**) TDP011 (carbon atom depicted in blue) is glutathionylated, but lacks interactions with Asp 113 and Phe39, but it does interact with other hydrophobic amino acids (not shown).

**Table 1 biomolecules-13-00461-t001:** Characteristics of major Nobo inhibitors.

Compound	EST	DMG	TDP011
Protein	DmNobo	AeNobo	DmNobo
PDB ID: ID	6KEP	7EBW	7DB3
Glutathionylation ^(a)^	NO	NO	YES
Interaction with Asp/Glu113	YES	YES	NO
Conformational change of Phe39	NO	NO	YES
IC_50_ (μM)	2.33 ± 0.08	0.293 ± 0.012	6.72 ± 1.48
Reference	Koiwai et al., 2020 [68]	Inaba et al., 2022 [69]	Koiwai et al., 2021 [74]

^(a)^ Inhibitors bound to GSH.

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
