# Peer review of "Compounds Inhibiting Noppera-bo, a Glutathione S-transferase Involved in Insect Ecdysteroid Biosynthesis: Novel Insect Growth Regulators"

_biomolecules, 2023, doi:10.3390/biom13030461_

Round 1
Reviewer 1 Report
This review summarizes the identification of chemical inhibitors targeting insect Nobo proteins that regulating insect molting by affecting ecdysteroid biosynthesis, and is very important for providing new strategies for pest control. This work is suitable for publication in Biomolecules. Before publication, I have two concerns as follows:
1. The English writing may need to be polished. For example, could the title for Figure 1 be changed as “A schematic illustration of the strategy of the identification of IGR against Nobo”?. In addition, because this article is a review, so the author should avoid the statement similar to a research article, like “Taken together, these results demonstrate that…” in line 170.
2. Nobo is a regulator of the biosynthesis of ecdysteroid. Given that some well-known chemical compounds, like juvenile hormone (JH), have been identified as an inhibitor of ecdysteroid biosynthesis, these molecules, especially JH, may be IGR candidates. Did the author analyze the effect of JH on Nobo activity? Alternatively, this point should be discussed in the section of Conclusion and future perspectives.
Author Response
This review summarizes the identification of chemical inhibitors targeting insect Nobo proteins that regulating insect molting by affecting ecdysteroid biosynthesis, and is very important for providing new strategies for pest control. This work is suitable for publication in Biomolecules.
Before publication, I have two concerns as follows:
1. The English writing may need to be polished. For example, could the title for Figure 1 be changed as “A schematic illustration of the strategy of the identification of IGR against Nobo”?. In addition, because this article is a review, so the author should avoid the statement similar to a research article, like “Taken together, these results demonstrate that…” in line 170.
Response: We have asked an experienced technical editor to check and polish the sentences throughout the revised manuscript, ensuring that the language is clear, concise, and consistent, and that the manuscript conforms to relevant style guidelines.
- Nobo is a regulator of the biosynthesis of ecdysteroid. Given that some well-known chemical compounds, like juvenile hormone (JH), have been identified as an inhibitor of ecdysteroid biosynthesis, these molecules, especially JH, may be IGR candidates. Did the author analyze the effect of JH on Nobo activity? Alternatively, this point should be discussed in the section of Conclusion and future perspectives.
Response: We have followed the reviewer's suggestion and included the discussion that was requested (L331-338 of the revised manuscript).
Reviewer 2 Report
This review summarises the recent findings for identifying and characterizing several chemical compounds that inhibit the Nobo protein in Drosophila melanogaster and Aedes aegypti as search for potential inhibitors.
While the review is comprehensive, I found it very descriptive and it mainly introduces the work of the author's group. There are a lot of descriptions such as 'We investigated', 'To identify AeNobo inhibitors, we performed', 'Next, we investigated' in this review, and make the paper looks like a conference presentation but not a review article. I would suggest that the authors revise the manuscript focus on making the review more critical where possible.
In addition, your review lacks analysis and hindsight on the subject that is reviewed. A review is not just to categorize what have been done, but most importantly, to critically analyze the information, synthesize the results from the literature, and provide fresh and insightful argument and/or discussion on the published results. Moreover, only 38 references were cited, poor information was provided, and the depth of the review is not enough, which makes this review lack of enough readers' interest.
The text is very fluid and well written in the introduction but the rest of the text needs to well organized and rewritten.
Line 76: The glutathione S-transferases (GSTs) family not 'glutathione S-transferase (GST) family'.
Line 76-78: Requires the incorporation of literature on the distribution and function of GSTs, the following references are suggested: Biochem. J. 2001, 360: 1−16; J. Agric Food Chem, 2022, 70: 2265−2279.
Line 89-91: Also lack of citations when statement the expression and purification of soluble recombinant proteins using the Escherichia coli expression system. The following references are suggested: Biochem. J. 1997, 324: 97–102; Pest Manag. Sci. 2011, 68: 764–772; Appl. Microbiol. Biotechnol. 2014, 98: 8947–8962.
Line 126: Confirm if there is lack of a space between 17 and β-Estradiol.
Author Response
This review summarises the recent findings for identifying and characterizing several chemical compounds that inhibit the Nobo protein in Drosophila melanogaster and Aedes aegypti as search for potential inhibitors. While the review is comprehensive, I found it very descriptive and it mainly introduces the work of the author’s group. There are a lot of descriptions such as ‘We investigated’, ‘To identify AeNobo inhibitors, we performed’, ‘Next, we investigated’ in this review, and make the paper looks like a conference presentation but not a review article. I would suggest that the authors revise the manuscript focus on making the review more critical where possible.
Response: We have made changes to the sentences in response to the reviewer's request to the greatest extent possible. However, we would like to note that several sentences still use "we" as the objective pronoun.
In addition, your review lacks analysis and hindsight on the subject that is reviewed. A review is not just to categorize what have been done, but most importantly, to critically analyze the information, synthesize the results from the literature, and provide fresh and insightful argument and/or discussion on the published results. Moreover, only 38 references were cited, poor information was provided, and the depth of the review is not enough, which makes this review lack of enough readers’ interest.
Response: We have improved the content of the manuscript as much as we can. We have also included more citations. In the revised manuscript, there are 79 references.
The text is very fluid and well written in the introduction but the rest of the text needs to well organized and rewritten.
Line 76: The glutathione S-transferases (GSTs) family not ‘glutathione S-transferase (GST) family’.
Response: In the revised manuscript, we have described “the GST family” (L117 of the revised manuscript).
Line 76-78: Requires the incorporation of literature on the distribution and function of GSTs, the following references are suggested: Biochem. J. 2001, 360: 1−16; J. Agric Food Chem, 2022, 70: 2265−2279.
Response: We have cited the references (L39 of the revised manuscript).
Line 89-91: Also lack of citations when statement the expression and purification of soluble recombinant proteins using the Escherichia coli expression system. The following references are suggested: Biochem. J. 1997, 324: 97–102; Pest Manag. Sci. 2011, 68: 764–772; Appl. Microbiol. Biotechnol. 2014, 98: 8947–8962.
Response: We have cited the suggested references (L133 of the revised manuscript).
Line 126: Confirm if there is lack of a space between 17 and β-Estradiol.
Response: We have confirmed that no space between “17” and “β-Estradiol” is needed. Therefore, we have still kept the original description (L171 of the revised manuscript).
Round 2
Reviewer 2 Report
The authors have addressed all my concerns and I recommend this manuscript to be accepted for publication in Biomolecules.